# Behavioural Systems Mapping of Solid Waste Management in Kisumu, Kenya, to Understand the Role of Behaviour in a Health and Sustainability Problem

**DOI:** 10.3390/bs15020133

**Published:** 2025-01-26

**Authors:** Joanna Davan Wetton, Micaela Santilli, Hellen Gitau, Kanyiva Muindi, Nici Zimmermann, Susan Michie, Michael Davies

**Affiliations:** 1Centre for Behaviour Change, Department of Clinical, Educational and Health Psychology, University College London, London WC1E 7HB, UK; micaela.santilli.21@ucl.ac.uk (M.S.); s.michie@ucl.ac.uk (S.M.); 2African Population and Health Research Center, P.O. Box 10787, Nairobi 00100, Kenya; hgitau@aphrc.org (H.G.); kmuindi@aphrc.org (K.M.); 3Institute for Environmental Design and Engineering, The Bartlett Faculty of the Built Environment, University College London, London WC1H 0NN, UK; n.zimmermann@ucl.ac.uk (N.Z.); michael.davies@ucl.ac.uk (M.D.)

**Keywords:** behaviour change, COM-B model, behaviour change wheel, systems thinking, behavioural systems mapping, waste management, Kenya

## Abstract

Poor solid waste management in Kisumu, Kenya, contributes to adverse health, social, and environmental outcomes as a result of open burning, illegal dumping, and reliance on landfills. Taking Kisumu as a case study, we use behavioural systems mapping (BSM) for the purpose of understanding the role of behaviour in this complex problem. We qualitatively analysed transcripts from focus groups and interviews with 45 stakeholders in Kisumu to construct a BSM of the perceived actors, behaviours, and behavioural influences affecting waste management, as well as causal links. Influences were analysed using the capability, opportunity, and motivation model of behaviour (COM-B). The resulting BSM connects 24 behaviours by 12 different actors and 49 unique influences (30 related to opportunity, 16 to motivation, and 3 to capability). It reflects three sub-systems: policy-making, public waste management, and the policy–public interface. Six key feedback loops are described, which suggest that cycles of underfunding are interlinked with problematic practices around the build-up, handling, and segregation of waste and conflicting public and political views around responsibility. We demonstrate how the BSM method can be used with transcript data and provide steps that others can follow to inform the design of systemic behaviour change interventions. Further research to validate and adapt this approach may extend the learnings to other countries and health and sustainability challenges.

## 1. Introduction

Global sustainability and health crises mean that governments urgently need to change a multitude of interconnected actions of citizens and organisations. In environmental policy contexts, behaviour change has often been equated with individual-level approaches and seen in contrast to systemic approaches. However, there is increasing recognition that effective environmental policies will need to be informed by an understanding of the causes and consequences of human behaviours in complex socio-technical systems. Although systems mapping and modelling approaches are widely used in the development of national and local environmental policies, few approaches are designed to represent the role of people’s behaviours, and few are linked to frameworks for designing behaviour change interventions. Behavioural systems mapping is a recent approach developed in connection with the Behaviour Change Wheel framework ([61]) for the purposes of understanding and changing human behaviour in complex systems and for informing policy decisions ([3]; [42]; [88]). In this paper, we apply behavioural systems mapping to the major challenge of municipal solid waste management (MSWM) in Kisumu County, Kenya.

### 1.1. The Problem of Municipal Solid Waste Management

Globally, growing quantities of municipal solid waste contribute to adverse environmental, social and health outcomes ([18]; [47]; [85]; [89]). In low- and middle-income countries, the problem is exacerbated by the prevalence of open dumping and burning ([33]; [47]; [83]), low standards of formal waste management and reliance on informal waste picking ([5]; [33]; [89]). In sub-Saharan Africa, MSWM remains a major challenge despite past and ongoing strategies at local, national, regional and continental levels ([7]; [34]; [39]; [84]). The majority of waste ends up in controlled or uncontrolled landfill sites, where its decomposition leads to greenhouse gas emissions, contamination of soil and water, and associated diseases ([34]; [84]), which have the greatest impacts on vulnerable groups such as those living in poverty and in proximity to dumpsites ([6]; [58]; [72]).

This long-standing challenge is driven by a complex set of factors, including population growth, increasing urbanisation and economic development, in conjunction with a lack of municipal financial resources, infrastructure, technical expertise and management planning ([44]; [72]; [74]). It is also characterised by complex interactions between local government, private and informal waste sectors, and the public ([58]). Because of these complexities and its resistance to change, MSWM in LMICs has been described as a ‘wicked problem’ ([16]; [32]; [70]; [71]). This refers to problems that are ‘ill-defined, ambiguous, and contested, and feature multi-layered interdependencies and complex social dynamics’ ([79]) and that imply a need for systems thinking ([36]; [52]; [56]).

Despite understanding that urban waste management involves a complex interplay of human behaviours, existing systems research in low- and middle-income countries has mainly focused on modelling flows of waste itself ([75]; [76], [77]; [78]). An exception is research by [40] ([40]), who used a systems thinking approach to map the interconnected actions and actors involved in waste management in Kisumu, Kenya. Drawing on transcripts from participatory modelling activities, their analysis aimed to represent key actions and actors involved in the flow of waste materials to identify weak links that could be potential points for intervention. These included a mix of changes to citizen behaviours, such as reducing waste, sorting at source, and improving disposal practices; changes to actors’ roles, namely entrepreneurs and local governors; and changes to financial and market systems. This study demonstrates the feasibility and benefits of mapping actors and their actions within the waste system, although the authors noted that their simplified map may benefit from further analysis. One aspect that their analysis did not aim to capture is the range of influences on people’s behaviours within the system. Understanding influences on behaviour (including and in addition to the role of other people’s behaviours) is a key starting point in developing effective interventions to change behaviour ([22]; [41]; [61], [59]; [60]).

### 1.2. Behavioural Systems Mapping

Behavioural systems mapping (BSM) is a recent approach developed in the field of behavioural science for the purpose of helping to understand and change human behaviour in complex systems ([3]; [41]; [88]). It involves making explicit the actors, behaviours, and influences on behaviour within a system, as well as the nature of the relationships connecting these. The method may vary depending on the type of systems map being produced. For example, types of systems mapping that depict causal relationships may include (but are not limited to) connection circles ([62]), fuzzy cognitive mapping ([38]; [51]), and causal loop diagrams ([8]; [20]). Behavioural systems mapping is not intended to replace other systems mapping methods but aims to guide and specify what type of information should be usefully represented in the systems for understanding and changing behaviour. For example, it is slightly different from behavioural system dynamics, in which psychological and behavioural explanations of phenomena are incorporated into quantitative systems models ([10]; [53]), in that behavioural systems mapping is a qualitative approach that focuses on framing and analysing issues primarily in terms of the human behaviours involved.

A behavioural systems map is a tool to help understand what is causing people’s current behaviours and what needs to be altered to change behaviours and the system as a whole ([42]; [88]). This may be improved by linking influences in the map to existing theories and models of behaviour ([3]; [27], [28]; [42]). A useful model in this context is the capability, opportunity, and motivation model of behaviour (COM-B) ([61]), which proposes that these three conditions are necessary for any behaviour to occur (Figure 1). Capability, opportunity, and motivation form an interacting system with behaviour, making COM-B particularly compatible with systems mapping ([61]; [86]). The COM-B model forms the basis of the Behaviour Change Wheel framework, which can be used for designing interventions that address these behavioural influences ([61]).

In the initial development of the approach, BSM was used in a participatory process to address the decarbonisation of homes in Wales ([42]). While stakeholder participation is widely recognised as beneficial to the process and outcomes of systems mapping by increasing the validity and utility of maps ([4]; [9]), there are situations where this is impractical or inefficient. In such contexts, previous research demonstrates that it is feasible and valuable to derive systems maps from textual data, particularly transcripts of open or semi-structured interviews with stakeholders ([31]; [49]; [80], [81]). Several methods have been developed for deriving causal maps from transcripts ([31]; [49]; [81]; [90]). Key features of these approaches include identifying causal relationships in the initial steps of coding and maintaining an audit trail from the original transcripts to the final systems map, often facilitated by computer-aided qualitative data analysis software (CAQDAS).

### 1.3. The Present Study

In this study, we aimed to draw upon transcripts to build a BSM without the direct participation of stakeholders in the construction of the map and to detail our methodology so that others may benefit from this extension of the method. To do this, we focus on municipal waste management in Kisumu, Kenya, as a case study. Data were gathered through participatory focus groups, interviews, and workshops involving stakeholders from local government, industry, academia, community-based organisations, and residents’ associations, all aimed at exploring health and sustainability issues in Kisumu County. The data were collected as part of the Complex Urban Systems for Sustainability and Health (CUSSH) project ([25]; [63]), which led to two related studies focusing on waste management in Kisumu through the lenses of system dynamics and attention theory ([29]; [70]). This study complements the previous two by focusing on the systemic role of people’s behaviour in driving waste management challenges and linking this to a behavioural science framework.

Our research questions are as follows: (1) Which actors, behaviours, and influences on behaviour contribute to the current system of solid waste management in Kisumu? (2) What are the causal pathways and feedback loops connecting these behaviours and influences? We anticipate the outcomes of this study may help to advance the application of BSM using transcripts as a primary data source, as well as generating behavioural findings that can help to inform policy design and implementation through close collaboration between the CUSSH project and the Kisumu County Government.

## 2. Materials and Methods

### 2.1. Case Study: Solid Waste Management in Kisumu, Kenya

Kisumu County in Kenya can be seen as a case study for the waste management crisis in sub-Saharan Africa. Kisumu City, the county’s capital, is Kenya’s third largest city. Located in the West of the country on the edge of Lake Victoria, Kisumu forms an important national and international hub for transport and commerce. Nevertheless, at the time of our data collection in 2019, more than half of the county’s population was categorised as poor, and around half were estimated to live in informal settlements ([73]). The county’s population of around 1.1 million ([48]) was estimated to generate between 200 and 500 tonnes of solid waste per day ([29]; [68]; [72]). Up to 40% of waste is collected and, until 2022, was taken to the city’s main landfill, Kachok dumpsite, where it had accumulated since 1975 and was periodically burned to free up capacity ([6]; [29]). The majority of the county’s waste is openly burned or illegally dumped on vacant land, on roadsides, and in drainage channels, where it contributes to poor sanitation, disease and air pollution ([29]; [40]; [72]). This unsustainable situation will be further strained as the Kenyan population is expected to grow over the next few decades, accompanied by continued urbanisation and changing patterns of consumption ([29]; [65]; [72]).

Previous attempts to improve waste management in Kisumu have faced difficulties. Plans to establish a new landfill at a distance from the overflowing Kachok dumpsite were delayed for years, mainly because of resident resistance in the proposed locations ([6]; [29]). The Kisumu Integrated Solid Waste Management Plan, developed and revised by the county government, outlined a 10-year strategy from 2015 to 2025 that aimed to reduce waste at source, improve waste collection and recycling rates, and establish more environmentally sustainable methods of waste disposal ([21]). These outcomes were to be achieved through planning, investment in infrastructure, legal reforms, and public–private partnerships, as well as grassroots-level initiatives such as community clean-ups designed to build public capacity and willingness to participate in waste management. However, the strategy did not follow the original implementation schedule, and in 2019, it failed to bring about substantial changes ([6]; [29]; [72]).

### 2.2. Participants

Our data were transcripts from in-depth interviews and focus groups with 45 participants representing seven stakeholder groups, listed in Table 1. Participants were invited by the study researchers or a research partner in the Kisumu County Government with a purposive, snow balling approach to recruitment based on their expertise and familiarity with the waste management sector. We also included transcript data from a workshop of mixed stakeholders who were partners in a funding application for waste management in Kisumu. Workshop participants comprised partners from the CUSSH project, the Kisumu County Government and local universities (including some participants who also took part in the focus groups).

### 2.3. Data Collection

Focus groups and interviews were facilitated using a 3 h discussion guide written in English (Appendix A). This was developed for the wider programme of CUSSH research in Kisumu and therefore did not specifically address the present research questions. The discussion guide aimed to elicit a wide range of insights related to the CUSSH project’s aims around city health and sustainability goals (which were shared with participants), local decision-making, and the development and implementation of city interventions.

Focus groups and interviews were conducted over two days in July 2019 in a hotel venue near Milimani, Kisumu. Interviews were in English, with some clarifications and prompts in Kiswahili. The research team (J.D.W., H.G., K.M., and N.Z.) facilitated each discussion in pairs. All are women social scientists with postgraduate training and experience in conducting qualitative interviews for research. For focus groups, at least one Kenyan researcher facilitated each discussion and provided any necessary translation between English and Kiswahili. Because stakeholder group sizes varied between 1 and 10 (Table 1), discussions varied in length. Participants were offered regular breaks with refreshments. All discussions were audio recorded and transcribed in English.

A separate workshop for partners from the CUSSH project, the Kisumu County Government, and local universities was held at the same hotel venue prior to the interviews. The same researchers (J.D.W., H.G., K.M., and N.Z.) were present and contributed to the discussion. The meeting was held in English and focused on discussing a joint programme of research and interventions around solid waste management. The workshop was audio recorded and transcribed in English.

## 3. Analyses

The analysis followed nine steps, each outlined in more detail below:Familiarisation with transcripts;Qualitative coding of transcripts to identify actors, behaviours, influences on behaviour, and perceived causal relationships between these;Construction of an initial behavioural systems map;Expert review (round 1);Revision and simplification (round 1);Expert review (round 2);Revision and simplification (round 2);Selection of illustrative quotes and linking to the COM-B model of behaviour;Identification of causal pathways and feedback loops.

### 3.1. Step 1: Familiarisation with Transcripts

J.D.W. read each transcript and associated coding generated in another analysis ([70]) and produced researcher memos to capture reflections from reading the transcripts.

### 3.2. Step 2: Qualitative Coding of Transcripts

J.D.W. coded the transcripts inductively in NVivo 12 Pro software by adding new codes under pre-determined categories of ‘actor’, ‘behaviour’, ‘influence on behaviour’, or ‘relationship’ (Table 2). These are the component parts that build up a behavioural systems map. Descriptions were developed by two researchers (J.D.W. and S.M.) prior to coding.

Behaviour and influence codes were named in such a way that they could form variables in the behavioural systems map (i.e., could be said to increase or decrease). Relationships between variables were coded using the ‘relationship’ feature in NVivo. Each relationship was coded as ‘positive’ (same direction of change; as A goes up, B goes up and vice versa) or ‘negative’ (opposite direction of change; as A goes up, B goes down and vice versa). The NVivo relationships feature was also used to record which actors performed each behaviour.

### 3.3. Step 3: Construction of Initial Behavioural Systems Map

To select variables that were likely to be relevant and important to the system of waste management, three sets of criteria were examined for inclusion of variables in the behavioural systems map:(a)Code mentioned at least three times and in at least three transcripts;(b)Code mentioned at least three times and in at least two transcripts;(c)Code mentioned at least two times and in at least two transcripts.

Criterion A is the most conservative, while Criterion C is the least conservative. Criterion B was selected and applied as the best compromise. All relationships between the selected variables were included, as were the actors associated with each behaviour.

J.D.W. constructed the initial behavioural systems map as a diagram in Vensim PLE version 8.0 (https://vensim.com/), a system dynamics open-source software (Appendix A). Colours were used to distinguish behaviour and influence variables. Arrows were used to represent causal relationships, with ‘+’ and ‘−’ denoting positive and negative relationships.

### 3.4. Step 4: Expert Review (Round 1)

The initial behavioural systems map was separately reviewed by three pairs of researchers: those who collected the data (K.M. and G.H.); system dynamics experts from the CUSSH project, not involved in the data collection; and behaviour change experts who were not involved with the CUSSH project. Each pair of reviewers was sent a copy of the initial behavioural systems map, background information, and written instructions. Reviewers were asked to inspect the map and consider the following questions:To what extent does the diagram capture the most relevant and important information?Are there parts of the diagram that are too complex or too simple?Are there any variables or links missing?Are there any variables or links that are not needed?Are there any variables or links that seem incorrect?Do you have any other feedback points?

J.D.W. met with each pair of researchers and facilitated a semi-structured discussion based on the questions above. Each meeting was audio recorded and transcribed. Suggested changes to the map were categorised according to whether they related to the map in general or a specific part and whether the suggestion was to add, remove, or change variables or relationships.

### 3.5. Step 5: Revision and Simplification (Round 1)

J.D.W. re-read all transcripts and researcher memos and suggested changes to the map before making revisions and simplifications. The revised version of the map (round 1) was constructed in Vensim PLE version 8.0. Any variables that did not pertain to the existing system of solid waste management (i.e., pertained to an imagined ideal or future system) were removed. Some variables were re-named or combined. Then, the researcher made and recorded changes to the map suggested by the reviewers and the reasons for them. At this stage, based on the feedback received and re-reading of transcripts and memos, the map was organised into two sub-systems to aid readability, named *policy* and *waste ‘on the ground’* (Appendix A).

### 3.6. Step 6: Expert Review (Round 2)

The revised behavioural systems map was separately reviewed by two of the authors (S.M. and N.Z.) and an additional researcher from the CUSSH project. They were asked to provide feedback from the perspective of behaviour change (S.M.), system dynamics (N.Z.), or in-depth familiarity with the transcript data (additional researcher). Each was sent a copy of the revised map background information and asked to consider the questions above. J.D.W. met with each reviewer to discuss their feedback. Each meeting was audio recorded and transcribed, and suggested changes were recorded and categorised as before (see step 4).

### 3.7. Step 7: Revision and Simplification (Round 2)

The revised version of the map (round 2) was made using Kumu software (version 2). Kumu was chosen for its interactive features, which could increase the usability of the final map. As before, J.D.W. made and recorded changes to the map suggested by the reviewers. At this stage, based on the additional feedback received and to further aid readability, the two sub-systems in the map (see step 5) were re-organised into three sub-systems, named *policy-making*, *public waste management*, and *policy–public interface.*

### 3.8. Step 8: Selection of Illustrative Quotes and Linking to the COM-B Model of Behaviour

J.D.W. selected illustrative quotes from the transcript data that described each variable and provided context (Appendix A). These were included as an interactive feature in Kumu so that viewers can click on a variable to view the quotes. To link the map to the COM-B model of behaviour, influence variables were categorised into capability, opportunity, or motivation factors, and these were colour-coded in the map.

### 3.9. Step 9: Identification of Causal Pathways and Feedback Loops

J.D.W. identified the balancing and reinforcing feedback loops present in the map. This is a typical step in the analysis and interpretation of causal systems maps, particularly causal loop diagrams because feedback loops can underlie important dynamics of a system. In a reinforcing feedback loop, there is a ‘snowball effect’ whereby a change in one variable ultimately leads to more change in that variable in the same direction, reinforcing the change. In a balancing feedback loop, change in a variable instead leads to a self-correcting effect, which limits runaway increase or decrease. In the Results section, we do not exhaustively describe every feedback loop in the map; short loops of two or three variables can be spotted relatively easily, but we focus on more complex loops, which may be harder to identify in the map and less obvious as potential drivers of the issue.

## 4. Results

Figure 2 shows the behavioural systems map of solid waste management in Kisumu. We recommend viewing this map interactively at https://kumu.io/JoHale/waste-in-kisumu (accessed 1 October 2024), where it is possible to zoom in and out and click on variables for illustrative quotes.

The overall map is organised into three sub-systems, which were identified through expert review and serve to help navigate the map by grouping the variables into related domains. The three sub-systems are *policy-making*, *public waste management*, and the *policy–public interface*. The *policy-making* sub-system describes processes of developing and implementing waste policy involving the county government, county assembly, national government, NGOs, and others. The *public waste management* sub-system describes practices happening ‘on the ground’, such as generating, segregating, mixing, collecting, transporting, picking, and burning waste. The *policy–public interface* sub-system connects these two sub-systems and describes the interplay between policy consultation and transparency and public attitudes and motivations.

### 4.1. Which Actors, Behaviours, and Influences on Behaviour Contribute to the Current System of Solid Waste Management in Kisumu?

In Table 3, we list the behaviours within each sub-system of the behavioural systems map and the actor(s) connected to them. Illustrative quotes describing each behaviour are provided in Appendix A, or they can be viewed by clicking on each behaviour in the interactive online map.

Table 4 lists the influences on behaviour, categorised according to the COM-B model into capability, opportunity, and motivation variables. These can be viewed in the interactive online map by clicking ‘show COM-B’. Illustrative quotes describing each influence are provided in Appendix A, or they can be viewed by clicking on each influence in the interactive online map.

Only three variables related to people’s capability (knowledge and skills). In other words, stakeholders did not often point to people’s abilities as the main influences shaping people’s current actions. The majority of influences described in the transcripts are related to opportunity (features of people’s physical and social environments), followed by motivational influences (automatic or reflective mental processes that drive behaviour). This is consistent with the nature of Kisumu’s waste management situation as a highly systemic problem made up of many interactions between different actors and local sites and infrastructure.

### 4.2. What Are the Causal Pathways and Feedback Loops Connecting These Behaviours and Influences?

We next describe the main causal pathways and feedback loops within each sub-system of the map and the connections between sub-systems. Not all possible pathways and loops are described exhaustively; we focus on those that appear to explain important patterns within and between the sub-systems and be relevant to add to stakeholders’ knowledge (as expressed in the transcript data) of the waste management challenges in Kisumu.

#### 4.2.1. Policy-Making Sub-System

Figure 3 shows the policy-making sub-system. Within this sub-system, there are two main reinforcing loops. The first reinforcing loop (‘Loop 1’) involves the behaviour *allocating county budget to waste* by county government employees and county assembly members. It represents that the less county budget allocated to waste, the less *county budget for waste management* and the fewer *human resources*. Human resources also depend on *government partnership with private waste collectors* (forming a balancing loop) and influence *enforcing waste management policy.* With fewer human resources, the greater the *government belief that the public should participate in waste management*, and the less *government sense of responsibility for waste*. The less sense of responsibility, the less *political prioritisation of waste management* and the less allocating of budget, completing the loop. *Departmental competition* was also described as an influence on county budget for waste management.

The second main reinforcing loop (‘Loop 2’) connects behaviours by county government employees and others of *gathering evidence*, *developing waste policy, implementing waste initiatives, and monitoring and evaluating policy initiatives*. There are several external influences on these behaviours, including scientific information, expertise in waste management, statutory environmental impact assessments, and delays in receiving resources.

The two feedback loops are connected in several ways. First, county budget for waste management and gathering evidence forms a balancing loop, whereby (for instance) more budget allows more gathering of evidence, which depletes the budget. Secondly, implementing waste initiatives depends on county assembly members *approving waste policy*, which is influenced by the political prioritisation of waste management. Thirdly, *visibility of positive impacts of waste management* influences both political prioritisation and also *external funding for sustainability initiatives*, which then influences implementation.

#### 4.2.2. Public Waste Management Sub-System

Figure 4 shows the public waste management sub-system. Two main sources of waste are represented in this sub-system. The first is *generating waste at market stalls*. As a consequence, a *build-up of waste at markets* influences market traders and residents *taking waste to bins and skips*. The *availability of bins and skips* influences taking waste to bins and skips and segregating waste there. The second source of waste is residents *generating waste at home*. A build-up of waste in the home leads to *burning waste, composting*, and *paying for waste collection*. Build-up of waste in the home and at markets both lead to *scattering waste*, which is influenced by an *existing habit to scatter waste*, affected by *sense of ownership of public space*.

As waste builds up in bins and skips, in homes, and scattered in public, this leads to collecting waste and transporting it to the dumpsite. Collecting waste is influenced by the *number of waste collection trucks*. This is reflected in the actors depicted as involved in waste collection (which are not necessarily exhaustive): *collecting waste from bins and skips* is performed by county government employees and waste pickers; *collecting residential waste* is performed by private waste collectors; and *collecting scattered waste* is performed by waste pickers and residents’ association members.

*Transporting waste to the dumpsite* is part of several feedback loops that we will describe here. In Loop 3, *transporting waste to the dumpsite* influences a build-up of waste there. This leads to *picking waste materials to reuse*, which influences *opportunity to make income from reused or recycled waste*, in turn influencing *poverty*, which influences *paying for waste collection,* and this feeds back to *collecting residential waste* and *transporting waste to dumpsite*. However, the opportunity to make income also depends on the *value of materials recovered from waste* for recycling. This is part of Loop 4: transporting waste to the dumpsite leads to *mixing previously segregated waste* (within the same truck or upon entering the dumpsite), and with more mixing, the less recycling can be carried out. In Loop 5, mixing previously segregated waste also leads to *uncertainty about where waste ends up*, which influences *public motivation to segregate waste for collection*, which in turn influences the segregating of waste at bins and skips and at home.

#### 4.2.3. Policy–Public Interface Sub-System

Figure 5 shows the policy–public interface sub-system. This sub-system includes one behaviour, *public capacity building*, by national and county government employees, NGOs, academics, and residents’ association members. This includes activities such as community education and clean-up days, which were described as having the purpose of increasing aspects of public capability, opportunity, and/or motivation to engage in waste management.

Public capacity building is part of a causal pathway that connects the policy-making sub-system to the public waste management sub-system. Capacity building follows from implementing waste management initiatives and is influenced by the requirement for the government to undertake *statutory community consultation*. Public capacity building positively influences *public knowledge*, *awareness, and attitudes about participating in waste management,* which in turn positively influence *public motivation to participate in waste management* and to segregate waste for collection.

The policy–public interface sub-system contains several other direct and indirect influences on public motivation to participate in waste management. These include *public trust in county government*, which is influenced by *government transparency*, and *public sense of responsibility for waste*, influenced by the *belief that waste management is the government’s responsibility*. This belief is influenced by *stigma attached to handling waste*, which also influences the positivity of attitudes about participating in waste management.

#### 4.2.4. Connections Between the Sub-Systems

We have already described how public capacity building is part of a causal pathway that connects from the policy-making sub-system to the public waste management sub-system. Here, we describe other relationships and feedback loops that connect behaviours within different sub-systems.

In the policy-making sub-system, allocating county budget for waste management influences the *availability of bins and skips* and the *number of waste collection trucks*, which affect behaviours in the public waste management sub-system. The number of trucks influences *collecting scattered waste* and in turn the *build-up of scattered waste*. With more build-up of scattered waste comes greater political prioritisation of waste management (or vice versa), which influences allocating county budget and thus forms a feedback loop (Loop 6). Other influences from the policy-making sub-system to the public waste management sub-system include *enforcing waste policy*, which negatively influences *scattering waste* and *schemes to subsidise the cost of recycling*, which positively influence the *opportunity to make income from reused or recycled waste*.

From the public waste management sub-system, a build-up of waste in the dumpsite was identified to positively influence *plans for relocating the dumpsite*. As waste built up in the past, this led to more intense plans for relocation. Relocation of the dumpsite could ease pressure on the need to implement (other) waste management initiatives, which is part of the policy-making sub-system.

## 5. Discussion

This study aimed to use the case study of municipal waste management in Kisumu, Kenya, to develop a behavioural systems map (BSM) representing the actors, behaviours, and influences on behaviour that contribute to the county’s waste challenges and to analyse the behavioural influences using the COM-B model of behaviour. We also aimed to demonstrate the usability of transcript data to build a BSM and document our methodology so that others may build on it. Our step-by-step analysis involved qualitatively coding transcripts to thoroughly identify the behavioural components that stakeholders described as contributing to Kisumu’s waste management issues, as well as relationships among these, and transferring this information into a BSM that was refined through two documented rounds of expert review. The resulting BSM allowed us to identify causal pathways and feedback loops connecting the behaviours of multiple actors in the public and policy spheres. Similar to Gutberlet et al.’s previous systems mapping study of household waste management in Kisumu ([40]), we depict a complex system involving many of the same actors at county, city, business, and household levels. However, whereas Gutberlet et al. focus predominantly on actions that happen to waste (or products before they become waste), our data led us to also map a range of policy-making behaviours.

The overall behavioural system of MSWM in Kisumu can be viewed in three sections, or ‘sub-systems’. The first relates to local policy-making. It encompasses the roles of government employees, assembly members, non-governmental organisations (NGOs), and academics in evidence-gathering, policy development and approval, implementation, and monitoring. Feedback loops in this part of the BSM suggest that the government’s lack of human resources reinforces, and is reinforced by, a lack of political prioritisation for waste management in budgets (in agreement with [72]), which is motivated by the view that the public should participate to meet the shortfall. This problematic cycle also hampers a more virtuous policy cycle of developing, implementing, monitoring, and evaluating waste policy. The more money diverted to gathering evidence, the less is available for human resources. While previous studies have found that budgetary and human resource constraints are barriers to effective MSWM in Kenyan and other sub-Saharan African cities ([35]; [47]; [72]; [82]; [91]), our study illustrates how these factors can be tied up in cycles of behaviours that perpetuate a lack of resources for implementing evidence-based policies.

The second section of the BSM pertains to the actions of residents, businesses, traders, and civil groups in the processes of generating waste—particularly at homes and markets—and then dealing with it through more or less desirable routes. This section includes considerable overlap with Gutberlet et al.’s map ([40]), specifically actions such as generating, composting, burning, sorting, collecting, transporting, dumping, and picking/scavenging waste, although our BSM goes into less granular detail and includes influences on the beahviours. Feedback loops in this part of the BSM suggest that public motivation to segregate waste (which would be desirable for recycling and recovering value) is undermined ([70]) by the practice of mixing any waste that was segregated at bins or skips when these are emptied into trucks to take the waste to the dumpsite. This occurs because there are too few bins, skips, and trucks (due to the lack of waste budget) to operate a preferable system of keeping waste streams segregated. Consequentially, there are fewer opportunities for government, businesses, or residents to recycle materials or recover income from waste. This is important for the promotion of a more circular economy, which has not only environmental but also socio-economic benefits through the creation of livelihoods ([11]; [26]). Lack of political will, funding, and public awareness have been previously identified as barriers to circular waste management in sub-Saharan Africa ([26]; [64]); our findings add that undermining public motivation may play a role.

The third section of the BSM describes some of the interface between policy-makers and the public, which is characterised by capacity-building efforts, such as community education and clean-up events. While these efforts are designed to increase the capability, opportunity, and motivation of residents to engage with waste management, two critical factors could influence their success: first, public trust in the county government (linked to government transparency around budgets and accountability), and second, the public’s belief that waste management is the government’s responsibility. Disagreement over government and public responsibility for waste management in Kisumu was prominent in our transcript data ([70]). This aligns with Gutberlet et al.’s finding that the responsibilities of the city/county are a point of confusion that needs to be addressed ([40]). Our findings add that the public’s viewpoint may partly stem from the undesirability of handling waste, while the government’s viewpoint relates to their lack of budget to deal with it.

A key contribution of this study is the use of the COM-B model to analyse the influences on behaviour represented in the BSM ([42]). This takes us further than previous studies in understanding the *types* of factors that drive behaviour in the waste management system. Capability factors (knowledge and skill) were only seen to play a small role. Most influences on behaviour in our map related to opportunities afforded by the physical environment, such as waste infrastructure and financial resources, and (to a lesser extent) social influences, such as the social stigma around waste handling. A range of motivational factors were also seen to play a role, particularly actors’ (often opposing) beliefs and attitudes surrounding waste management and sense of responsibility, as well as plans and existing habits.

### 5.1. Practical and Scientific Implications

Our findings have practical implications for MSWM intervention in Kisumu and places facing similar waste issues. Where there is a mismatch between government and public attitudes about who should take responsibility for waste management, it may be important to facilitate dialogues between government officials and community representatives to address the disagreement and develop a collaborative approach. The CUSSH project has subsequently helped facilitate such dialogues through a programme of public engagement between residents, local policy-makers, journalists, and other stakeholders ([50]). Increasing the visibility of positive impacts of waste management, e.g., through campaigns highlighting health and sustainability benefits, could also help to increase both political prioritisation and external investment of funding ([45]). In addition to this, increasing the transparency of government decision-making and expenditure could help to build public trust ([2]) and avoid undermining public capacity-building efforts that are already in place (although previous research suggests this may not be guaranteed ([37])). Efforts could be improved by shifting the emphasis away from informing, educating, and persuading residents to designing behaviour change interventions that address the physical opportunity barriers they are also facing ([12]; [30]; [66]), such as limited availability of waste receptacles, unaffordable home waste collections ([72]), and landlord absenteeism. This is not only relevant for Kisumu and locations in the global south but applies just as much to global north cities where there is also over-reliance on informational approaches to reduce littering ([17]).

Scientifically, this study contributes to the new and growing literature on behavioural systems mapping ([3]; [15]; [23]; [24]; [27], [28]; [42]; [57]; [88]; [92]). The main feature of behavioural systems mapping (relative to other systems mapping approaches) is that it makes explicit the role of specific actors, behaviours, and influences on behaviour and how these are connected in a system ([42]). This approach offers much more nuance than traditional qualitative studies of barriers and enablers to behaviour change (which usually list factors in relation to one ‘target’ behaviour), while still enabling the use of frameworks such as the Behaviour Change Wheel (BCW) to aid in identifying suitable types of intervention. Previously published BSM studies have involved participatory stakeholder workshops and interviews ([3]; [15]; [42]; [92]), although two projects have drawn upon wider data sources, including literature reviews ([3]; [92]). Training in BSM led by the UCL Centre for Behaviour Change and other institutions has also focused primarily on group model-building approaches. Our study demonstrates the feasibility of constructing BSMs from transcript data when access to stakeholders is no longer possible and provides a detailed step-by-step approach that others can follow and adapt.

### 5.2. Strengths, Limitations, and Future Research

A strength of this study is the application of behavioural theory in building and interpreting the BSM. Many systems science studies of waste management ([79]; [36]; [56]; [29]) and other environmental and health issues model material stocks and flows and treat policies or behaviours as extraneous variables. Studies that do incorporate human behaviour in systems models rarely link to theories about the determinants of behaviour. This makes it hard to connect insights from the systems map/model to interventions that could change behaviour. In our study, the use of the COM-B model bridges that gap by linking the components of the map to a coherent intervention design framework, the BCW. Additional research could build on the present study by using the BCW to explore potential intervention options in detail with local stakeholders ([87]).

Another strength of this study is that it draws from a wide range of stakeholder perspectives ([70]), providing different views of MSWM issues ‘on the ground’. Forty-five people from varied roles in local government, industry, academia, and community organisations took part in the discussions that formed our dataset, providing rich insight into waste issues and how these connected with the wider context of climate and health policy-making in Kisumu. Most of the stakeholders (with the exception of industry) took part in group discussions, which can reduce reliance on individual views. Combining such a range of perspectives is a key tenet of systems thinking approaches, and it can lend fairness, accuracy, and legitimacy to resulting systems maps ([4]; [42]).

These benefits are more fully realised when stakeholders take part in the mapping process itself ([4]), and a limitation of the study is that we could not give participants that opportunity due to project constraints during the COVID-19 pandemic. We mitigated this partly through two rounds of expert review by behavioural scientists, systems scientists, and local researchers to improve the quality of the BSM. However, a valuable next step would be to engage local stakeholders in interpreting, challenging, and using the BSM as a tool for building joint understandings. This could also present an opportunity to include any additional perspectives that could have been missing among those interviewed. Our inclusion of interactive features in the online version of the BSM (e.g., to view sub-systems, toggle COM-B colour-coding, and click on elements to read illustrative quotes from the transcripts) could help stakeholders engage with the map.

The successful use of transcript data to create a BSM is also a strength of the study. We were able to reuse data collected for other purposes within the CUSSH project to carry out a rich behavioural analysis that complements two other studies on the same issue ([29]; [70]). Re-analysis of data is not only good for scientific efficiency ([69]) but can offer benefits through the opportunity to triangulate insights gained through different methods.

However, compared with group model-building workshops, our qualitative coding of transcripts was a very time-intensive method. This limitation has been acknowledged in previous studies ([31]; [49]; [90]) and could present a barrier to following our steps. Part of the reason was that our transcripts were very long and not specifically directed by the research questions. The analysis could be quicker by collecting or editing transcript data to be more focused. By increasing access to and knowledge of AI-assisted qualitative analysis ([14]; [19]; [43]; [54]), made possible by recent large language models and generative AI, future research could potentially make use of these tools to increase the efficiency of coding and systems modelling ([55]).

Finally, it is important to note that the data analysed in the present study were collected in 2019, and in the time elapsed, there will have been changes to MSWM in Kisumu. One of the most significant changes has been the relocation of the Kachok dumpsite ([72]). Work began in 2021 to establish a new dumpsite in the Muhoroni sub-county and decommission the old site ([13]; [67]), with plans to convert the Kachok site into a recreational garden, although this has not yet been realised ([46]). This and other developments could alter the picture of MSWM in our BSM, underscoring the relevance of further work to engage local stakeholders in interpreting, challenging, and using the map. Nevertheless, the issues discussed reflected long-term patterns of behaviour that have persisted for decades and continue to hinder solid waste management in sub-Saharan African contexts ([1]; [7]; [26]). Thus, recent and future developments in Kisumu are unlikely to alter all the lessons learned from this case study and do not detract from the wider value of the project in advancing BSM.

This map was specifically developed to understand the MSWM in Kisumu. We recognise that the behaviours, influences, and actors identified in this map may vary in other nations or different cities within Kenya. While some general trends may be applicable to other sub-Saharan African cities, we recommend that this map be validated and adapted for use in different contexts to ensure its relevance and accuracy in other case scenarios.

## 6. Conclusions

This study developed a behavioural systems map (BSM) to represent the actors, behaviours, and influences contributing to municipal solid waste management challenges in Kisumu, Kenya. Applying the COM-B model revealed that physical opportunity and motivational factors predominantly influence behaviour in this context. Through qualitative analysis of transcript data, we identified interconnected processes in local policy-making, public waste handling, and interactions between policy-makers and the public. The BSM revealed critical feedback loops that suggest that cycles of underfunding are interlinked with problematic practices around the build-up, handling, and segregation of waste and conflicting public and political views around responsibility.

Our findings suggest the need for collaborative dialogues between government and community representatives, increased transparency in decision-making, and interventions addressing physical opportunity and motivation barriers. Overcoming these barriers could help to promote more circular waste management practices, which can have additional benefits, such as generating income. This study contributes to behavioural systems mapping methodology by demonstrating the feasibility of constructing BSMs from transcript data. Future research and collaborative dialogues to improve MSWM could involve local stakeholders in interpreting and using BSMs, potentially employing AI-assisted analysis to enhance efficiency in BSM construction.

## Figures and Tables

**Figure 1 behavsci-15-00133-f001:**
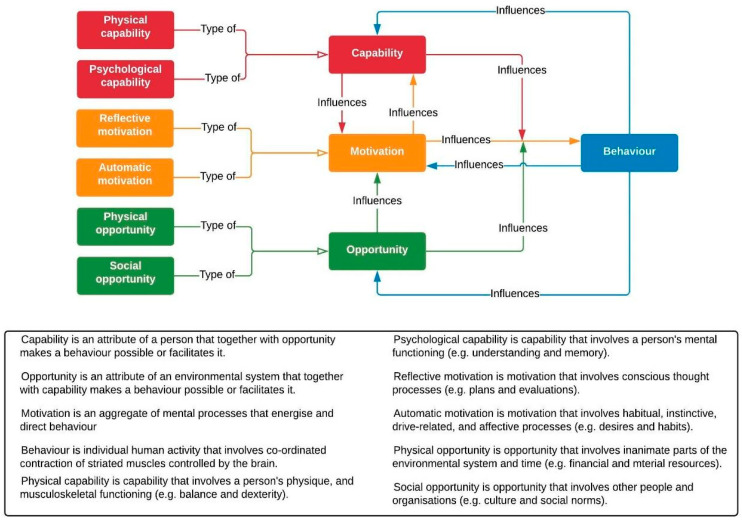
The COM-B model of behaviour. Reproduced with permission from [86] ([86]).

**Figure 2 behavsci-15-00133-f002:**
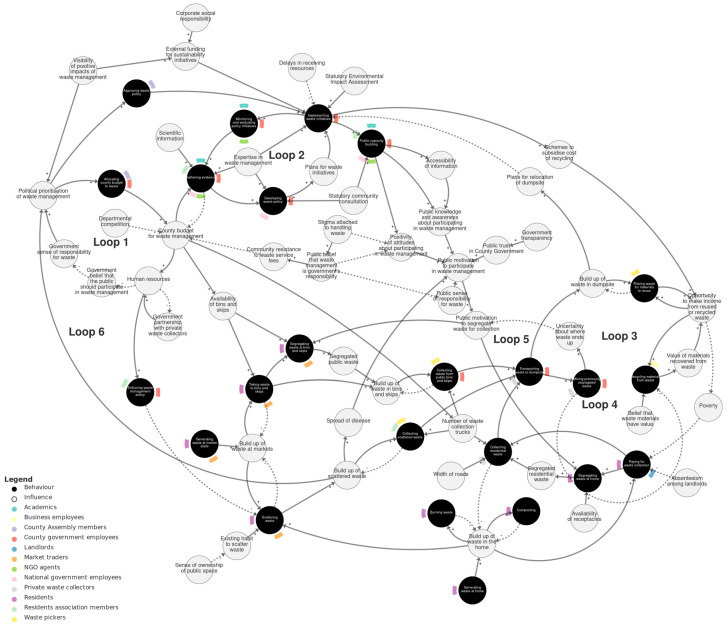
Behavioural systems map of solid waste management in Kisumu. Behaviours are shown as black circles. Actors (the people doing the behaviours) are shown as colour-coded flags around the edge of each behaviour. Influences on behaviour are shown as grey circles. Causal relationships are shown as arrows. Solid arrows are used for positive relationships (+). Dashed arrows are used for negative relationships (−). We recommend viewing this figure interactively online at https://kumu.io/JoHale/waste-in-kisumu (accessed on 1 October 2024).

**Figure 3 behavsci-15-00133-f003:**
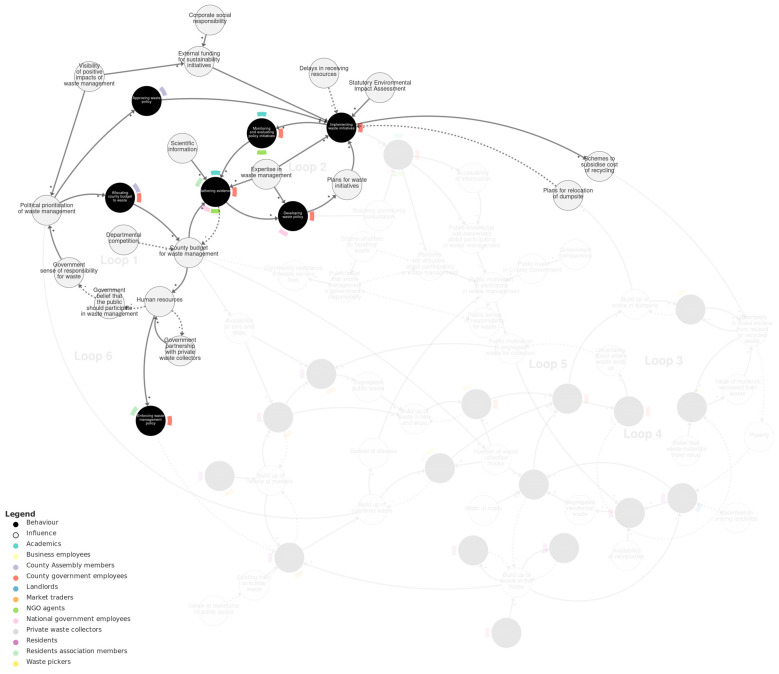
Policy-making sub-system. Loop 1 and Loop 2 are labelled with pale grey text. We recommend viewing this figure interactively online at https://kumu.io/JoHale/waste-in-kisumu (accessed on 1 October 2024).

**Figure 4 behavsci-15-00133-f004:**
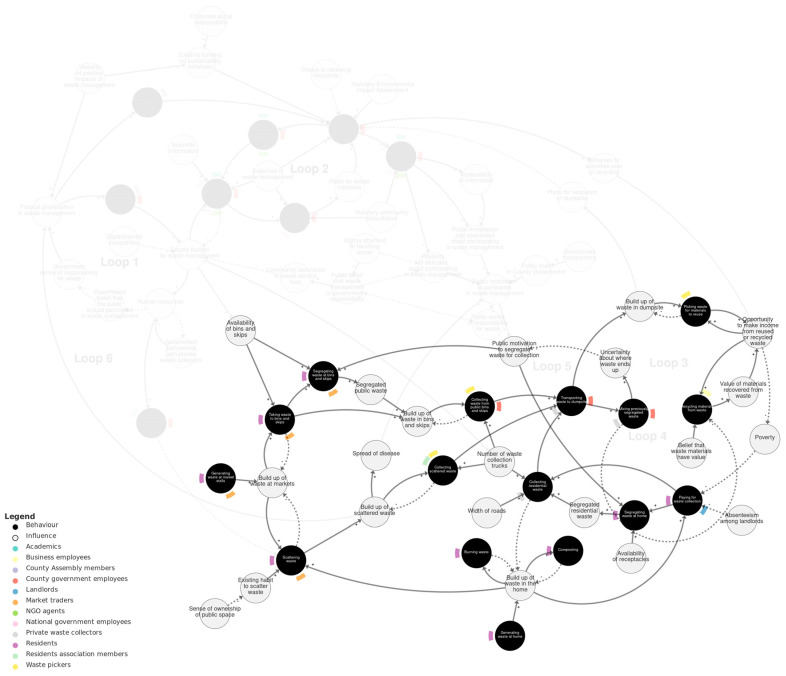
Public waste management sub-system. Loops 3–5 are labelled with pale grey text. We recommend viewing this figure interactively online at https://kumu.io/JoHale/waste-in-kisumu (accessed on 1 October 2024).

**Figure 5 behavsci-15-00133-f005:**
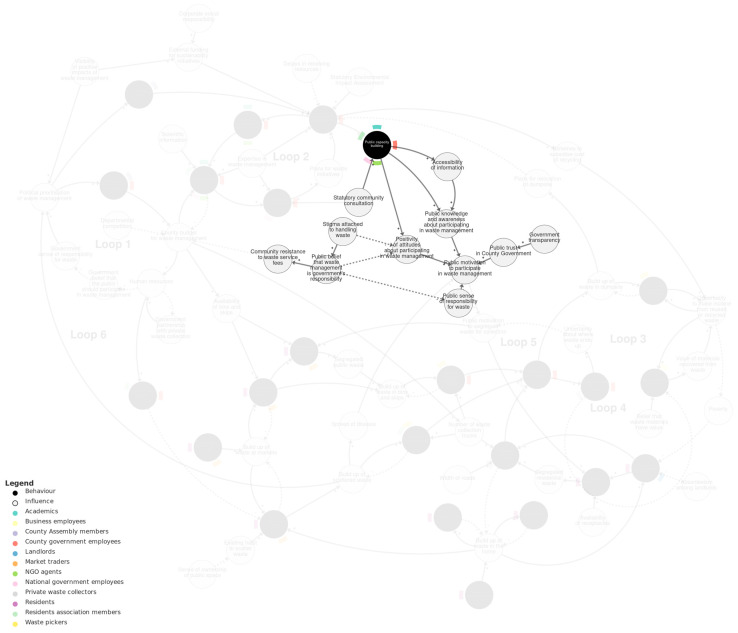
Public–policy interface sub-system. We recommend viewing this figure interactively online at https://kumu.io/JoHale/waste-in-kisumu (accessed on 1 October 2024).

**Table 1 behavsci-15-00133-t001:** Stakeholder groups interviewed.

Stakeholder Group	Number of Participants
Kisumu County climate change directorate	10
Kisumu County officials	8
Community-based organisations (CBOs) providing waste management and waste-to-energy services	9
Industrial associations of commerce and manufacturing	2
Academic researchers in sustainability, health, and waste management from local universities	8
Residents’ associations	7
Local factory-based industry	1

**Table 2 behavsci-15-00133-t002:** Coding categories.

Coding Category	Description
Actor	An individual, group, organisation, or sector that enacts a behaviour seen as relevant to the issue.
Behaviour	An action that is directly or indirectly observable.
Influence on behaviour	A variable that directly or indirectly affects the likelihood, frequency, or characteristics of a behaviour and is not itself a behaviour.
Relationship	An observed, perceived or expected causal relationship between behaviours and/or influences on behaviours, expressed explicitly or implicitly.

**Table 3 behavsci-15-00133-t003:** Behaviours and actors within three sub-systems of waste management in Kisumu.

Sub-System	Behaviour	Actor(s) Connected to the Behaviour
Policy-making	Allocating county budget to waste	County assembly members, county government employees
	Approving waste policy	County assembly members
	Developing waste policy	National government employees, county government employees
	Enforcing waste management policy	County government employees, residents association members
	Gathering evidence	Residents association members, academics, NGO agents, national government employees, county government employees
	Implementing waste initiatives	County government employees
	Monitoring and evaluating policy initiatives	Academics, NGO agents, county government employees
Public waste management	Burning waste	Residents
	Collecting residential waste	Private waste collectors
	Collecting scattered waste	Residents’ association members, waste pickers
	Collecting waste from public bins and skips	County government employees, waste pickers
	Composting	Residents
	Generating waste at home	Residents
	Generating waste at market stalls	Residents, market traders
	Mixing previously segregated waste	Private waste collectors, county government employees
	Paying for waste collection	Residents, landlords
	Picking waste for materials to reuse	Waste pickers
	Recycling materials from waste	Business employees
	Scattering waste	Residents, market traders
	Segregating waste at bins and skips	Residents, market traders
	Segregating waste at home	Residents
	Taking waste to bins and skips	Residents, market traders
	Transporting waste to dumpsite	Private waste collectors, county government employees
Policy–public interface	Public capacity building	Residents association members, academics, NGO agents, national government employees, county government employees

**Table 4 behavsci-15-00133-t004:** Influences on behaviour categorised according to COM-B.

Sub-System	COM-B Domain (*N* Influences)	Influence on Behaviour
Policy-making	Capability (1)	Expertise in waste management
	Motivation (6)	Corporate social responsibility
		Government belief that the public should participate in waste management
		Government sense of responsibility for waste
		Plans for relocation of dumpsite
		Plans for waste initiatives
		Political prioritisation of waste management
	Opportunity (10)	County budget for waste management
		Delays in receiving resources
		Departmental competition
		External funding for sustainability initiatives
		Government partnership with private waste collectors
		Human resources
		Schemes to subsidise cost of recycling
		Scientific information
		Statutory Environmental Impact Assessment
		Visibility of positive impacts of waste management
Public waste management	Capability (1)	Uncertainty about where waste ends up
	Motivation (5)	Belief that waste materials have value
		Existing habit to scatter waste
		Opportunity to make income from reused or recycled waste
		Public motivation to segregate waste for collection
		Sense of ownership of public space
	Opportunity (15)	Absenteeism among landlords
		Availability of bins and skips
		Availability of receptacles
		Build-up of scattered waste
		Build-up of waste at markets
		Build-up of waste in bins and skips
		Build-up of waste in dumpsite
		Build-up of waste in the home
		Number of waste collection trucks
		Poverty
		Segregated public waste
		Segregated residential waste
		Spread of disease
		Value of materials recovered from waste
		Width of roads
Policy–public interface	Capability (1)	Public knowledge and awareness about participating in waste management
	Motivation (5)	Positivity of attitudes about participating in waste management
		Public belief that waste management is government’s responsibility
		Public motivation to participate in waste management
		Public sense of responsibility for waste
		Public trust in county government
	Opportunity (5)	Accessibility of information
		Community resistance to waste service fees
		Government transparency
		Statutory community consultation
		Stigma attached to handling waste

## Data Availability

Data collected and generated by this research are considered confidential.

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
