# Peer review of "Behavioural Systems Mapping of Solid Waste Management in Kisumu, Kenya, to Understand the Role of Behaviour in a Health and Sustainability Problem"

_behavsci, 2025, doi:10.3390/bs15020133_

Round 1
Reviewer 1 Report
Comments and Suggestions for Authors
Is the problem described significant? Waste management is a very important problem occurring globally all over the world. The article emphasizes that improper waste management contributes to adverse health, social and environmental effects.
Is the article clear? What is the main question of the study? The article is clear. This paper addresses the following research questions: Which actors contribute to the current state of solid waste management in Kisumu? And what are the causal paths and feedback loops connecting the behaviors of these entities?
Is the question original and well defined? Are the results an improvement on current knowledge? The questions are rather obvious regarding waste management, and their reference to the analyzed area of ​​Kisumu is an important contribution to contemporary knowledge.
Does it have a scientific basis that allows you to test the hypotheses? The article does not present any research hypothesis(s).
Are the figures/tables appropriate? In general, the figures and tables are correct, but figures 2-5 are difficult to read due to the use of a very small font.
Are the results consistent with the content? And do they address the main question?
The results are consistent with the content, but the conclusions should be properly highlighted and what they mean for the analyzed entities should be shown.
Are the cited sources recent?
The cited sources are the latest and appropriately selected, but are largely based on the works of Asian authors.
I suggest including the following tips in the article:
1. The abstract should present the purpose and research questions.
2. The article presents various citation systems, please adapt the citations shown in lines 41-44, 103 to the system [number].
3. The data used in the article were collected in 2019, i.e. 5 years ago, so please justify whether they have not become outdated and whether the same answers would be provided now. All the more so because, as the Authors mention in lines 629-638, there have been significant changes in the analyzed area.
4. The same applies to the number of interviews conducted. Can any forward-looking conclusions be drawn based on interviews conducted with representatives of 45 entities, including 1 factory, for example? and present the mapping aspect.
5. The authors suggest that the presented system maps be viewed at: https://kumu.io/JoHale/waste-in-kisumu, where you can "enter" the interpretations of individual circles. But the question arises whether the map presented on the website is also authored by the authors of the article?, because the website does not mention any authors? On the other hand, the reader of the article does not have to want to browse the given page, he may want to draw conclusions based solely on the article... where figures 2-5 are difficult to read due to the small font.
6. In the discussion, the authors refer almost exclusively to the mapping method, and it would also be appropriate to get to the essence of the article, i.e. the authors should also supplement it with waste management.
7. The conclusions should point out what clearly results from the research for individual analyzed entities covered by the research.
8. In line 556 a reference is made to the CUSSH Kisumu website, rather this address should be included in the literature reference
Author Response
Thank you very much for your review report. We appreciate your feedback and suggestions to improve the manuscript. Please find our responses to specific comments below.
Comment 1: The abstract should present the purpose and research questions.
Response1: We have clarified the purpose and added research questions 1 and 2 to the abstract (lines 13-16).
Comment 2: The article presents various citation systems, please adapt the citations shown in lines 41-44, 103 to the system [number].
Response 2: We have updated the citations to the correct format (lines 44, 46, 106)
Comment 3: The data used in the article were collected in 2019, i.e. 5 years ago, so please justify whether they have not become outdated and whether the same answers would be provided now. All the more so because, as the Authors mention in lines 629-638, there have been significant changes in the analyzed area.
Response 3: Thank you for pointing this out. We have provided further justification in the discussion section that although the data were collected in 2019, the issues discussed reflected long-term patterns of behaviour that have persisted for decades and continue to hinder solid waste management in sub-Saharan African contexts (lines 645-647).
We have added an additional reference to a comprehensive Systematic Review and Meta-Analysis from 2023 to back up this justification. We have also added a recommendation that this map be validated and adapted for use in different contexts to ensure its relevance and accuracy in other case scenarios (lines 650-654).
Comment 4: The same applies to the number of interviews conducted. Can any forward-looking conclusions be drawn based on interviews conducted with representatives of 45 entities, including 1 factory, for example? and present the mapping aspect.
Response 4: We have provided further justification in the methods section that participants were recruited based on their expertise and familiarity with the waste management sector (line 178-179), and in the discussion section we have added that most of the stakeholders (with the exception of industry) took part in group discussions, which can reduce reliance on individual views (line 607-608). We would also note that the number and breadth of participants is comparable to other systems mapping studies.
Comment 5: The authors suggest that the presented system maps be viewed at: https://kumu.io/JoHale/waste-in-kisumu, where you can "enter" the interpretations of individual circles. But the question arises whether the map presented on the website is also authored by the authors of the article?, because the website does not mention any authors? On the other hand, the reader of the article does not have to want to browse the given page, he may want to draw conclusions based solely on the article... where figures 2-5 are difficult to read due to the small font. Response 5: We can confirm that the map at the Kumu website was created by the authors and we have added this information to the map at https://kumu.io/JoHale/waste-in-kisumu. We have increased the font size in Figures 2-5. Comment 6: In the discussion, the authors refer almost exclusively to the mapping method, and it would also be appropriate to get to the essence of the article, i.e. the authors should also supplement it with waste management. Response 6: Although we consider the systems mapping method the primary contribution of the article, we refer throughout the discussion section to the implications for waste management in Kisumu and more widely. We link our findings to existing studies of waste management and include a dedicated section on practical implications (line 556-575) as well as scientific implications. In the strengths and limitations we also discuss stakeholder involvement and the changing context of waste management in Kisumu since our data were collected. Thus we feel that there is an appropriate balance of discussion on the method and the topic of waste management, given the purpose and main contribution of our study. Comment 7: The conclusions should point out what clearly results from the research for individual analyzed entities covered by the research. Response 7: Our study did not seek to analyse individual participants or stakeholder groups, but rather to integrate their different perspectives so as to understand how their behavioural interactions contribute to waste management challenges in Kisumu. As such, the main conclusions apply to all of the entities involved and we cannot disentangle specific conclusions for specific entities. We discuss some implications for particular stakeholder groups earlier in the discussion section and have avoided repeating this in the conclusions so as not to add unnecessary length to the article. Comment 8: In line 556 a reference is made to the CUSSH Kisumu website, rather this address should be included in the literature reference.
Response 8: We have included the website as a reference.
Reviewer 2 Report
Comments and Suggestions for Authors
Behavioural systems mapping of solid waste management in Kisumu, Kenya
The article focuses on the behavioural systems mapping that used focus group data to understand solid waste management in Kisumu, Kenya. The suggestion was to understand systemic behavioral change interventions with the possibility that understanding behavioral change mapping might reduce waste.
Title: With solid waste impacting sustainability, health, and poverty, especially in a developing nation, the title could be more detailed to attract readers to the findings and to effect change. Kisumu, Kenya would probably have different materials in its solid waste than a suburban US neighborhood. Solid waste is based on income level, food, and weather. In a location where vegetables can be grown in a home garden, little food goes to waste and waste can be put into a compost pile to regenerate the soil. In a developing nation, if lower income people are given the opportunity to recycle materials, such as clothing, plastics, metal, or paper before it is contaminated by being mixed with waste, they could earn an income. Solid waste can be clean, such as electronics and clothing, or dirty, such as old food. Therefore, the term solid waste should be defined because one form of waste can replenish the earth and the other could be clean, recycled, and sold. I believe the authors have more details in their highly contributory article and could offer guidance to lessen solid waste’s negative impact on the world. As the article stands, they are just modestly saying they have developed and tested a behavioral mapping system that happens to focus on solid waste in Kenya and others might use their mapping system.
What about the title: “Behavioral systems mapping of solid waste management in Kisumu, Kenya: How a developing nation could separate clean plastics, clothing, metal and paper from food and provide lower-income citizens an income”
Below is an online article that showed details about recycling and composting in Kisumu:
https://www.kisumu.go.ke/battling-pollution-by-segregation-recycling-and-composting-of-solid-waste-at-ondiek-estate-in-kisumu/
“We used to collect waste and dump them at Kachok dumpsite or in our backyards but now through this project, we will do waste segregation at the household level. We have been supported with a bin and two gunny bags each to incentivize waste segregation. We are now working on a holding ground where we will sell the recyclables to dealers, and from the proceeds, we will plough some back to sustain the project while some will be re- invested in our welfare activities to generate more income. In future, we plan do recycling on our own and not sell to dealers” says Mr. Fred
The plastic waste will be sold to dealers at the waste recovery in Maendeleo market for recycling while the biodegradable/ organic will be sold to dealers at Obunga for composting hence reducing waste landfill.”
Abstract: The abstract now focuses on the behavioral systems mapping and the text includes different forms of the word “behavior” eleven times. The abstract should offer the reader background about Kisumu, Kenya, the income level, the types of solid waste (clean and dirty), and the current practices for reuse or disposal of the recyclables and compostables. With the income level, the reader will understand that one of the problems is poverty. If low-income residents have a way to make money from recycling clean waste before it is contaminated, that goes beyond just the article’s benefit of having tested a behavioural mapping system in Kisumu. The authors will have shown, by example, how application of the behavioural mapping system can result in a sustainable landscape, a healthy population, and income for lower income residents. The authors have findings that are in the Discussion that could be in the Results (lines 509 to 525). The authors have excellent findings on lines 325-326, under public waste management motivation and opportunity in Table 4, and lines 405 to 432 that could be discussed in the Abstract. This text gives the reader hope that there are solutions. The time has passed for academics to just offer a mapping system. The authors have far more detail in their article and should give this information to the reader. To really help, the authors need to be clear and offer simple suggestions for next steps that individuals in developing nations could take. With this information, others are more likely to also try the behavioural systems mapping.
Introduction: As with the abstract, the problem is not that the world has just needed to understand and change human behavior (lines 42 to 43) but, more specifically, that people in developing nations are poor and developing nations have few resources to wisely manage of all the components in solid waste to address sustainability, health, and poverty. If the reader is able to read the factors involved, they are more likely to see the value of understanding and changing human behavior.
On line 122 and 123, the text needs to be altered. The text states the authors were building a BSM without “the direct participation of stakeholders.” On lines 125 to 126, the authors discuss involving stakeholders by having focus groups, interviews, etc. It is hard to think of someone else being able to replicate the study without also doing their own interviews. Any data for secondary data analysis would have to have asked similar questions posed by the authors. Most generic population data gathering studies do not include questions about what citizens should do with recyclables and if they might earn an income.
The Figures 2, 3, 4 and 5 are small and would be hard for the lay person to understand and follow. Is it possible for the authors to offer the readers a very simple end-result systems map they could follow and implement? The color coding and interaction of the map are excellent (lines 610 to 612) but someone might only be able to read the article online or print the article. While the mapping is important, the authors have results from the interviews that could be implemented. Moving a dump site is worthwhile, but if the same waste is generated and taken to a new dump site, sustainability, health, and poverty will not be impacted.
Under limitations, the authors should write that this study involved individuals in Kisumu, Kenya and may not be generalizable to other communities, especially communities that are not in developing nations.
The authors have a worthwhile study and the hope is they can offer the reader more, because the authors have more.
Author Response
Thank you very much for your review report. We appreciate your feedback and suggestions to improve the manuscript. Please find our responses to specific comments below.
Comment 1: What about the title: “Behavioral systems mapping of solid waste management in Kisumu, Kenya: How a developing nation could separate clean plastics, clothing, metal and paper from food and provide lower-income citizens an income” Response 1: Thank you for the suggestions on the title. We have carefully chosen the title wording to reflect the aim and methodological contribution of the paper, whilst still being concise. We do not feel that the article sufficiently speaks to the separation of clean plastics, clothing, metal and paper from food, or how to provide lower-income citizens an income. Therefore we have not changed the title; however, we have added in the conclusions that overcoming the opportunity and motivation barriers we identified could help to promote more circular waste management practices which can have additional benefits such as generating income (lines 666-668). Comment 2: The abstract should offer the reader background about Kisumu, Kenya, the income level, the types of solid waste (clean and dirty), and the current practices for reuse or disposal of the recyclables and compostables. Response 2: We regret that due to word count constraints we are unable to add further context about Kisumu to the abstract. However, we have provided detailed background about Kisumu, the income level, types of waste and current practices in the materials and methods section (lines 146-173). Comment 3: The authors have findings that are in the Discussion that could be in the Results (lines 509 to 525). Response 3: In lines 509-512 we have clarified that these findings come from other studies. In lines 513-525 we are not presenting new results but rather we are interpreting this part of the map in relation to the previous research and explaining the implications. We feel this detailed interpretation is worthwhile in the discussion section because the results section may be quite complex for most readers. Comment 4: The authors have excellent findings on lines 325-326, under public waste management motivation and opportunity in Table 4, and lines 405 to 432 that could be discussed in the Abstract. Response 4: We have included the finding from lines 325-326 in the abstract. We regret that due to word count constraints we are unable to list the findings from Table 4 in the abstract. Unfortunately we also do not have space to include the feedback loops described in lines 405-432. Comment 5: The authors have far more detail in their article and should give this information to the reader. To really help, the authors need to be clear and offer simple suggestions for next steps that individuals in developing nations could take. With this information, others are more likely to also try the behavioural systems mapping. Response 5: Thank you for this suggestion. We have added to the conclusions that collaborative dialogues to improve MSWM could involve local stakeholders in interpreting and using behavioural systems maps (line 670-671). Considering the limitations of our study and that the map has not yet been validated with stakeholders, we are hesitant to offer steps that developing nations could take, as these would be highly context-dependent. We have included a paragraph about this at the end of the limitations (lines 650-654). Comment 6: On line 122 and 123, the text needs to be altered. The text states the authors were building a BSM without “the direct participation of stakeholders.” On lines 125 to 126, the authors discuss involving stakeholders by having focus groups, interviews, etc. Response 6: We have amended line 122-123 to clarify that we meant without the direct participation of stakeholders in the construction of the map. Comment 7: The Figures 2, 3, 4 and 5 are small and would be hard for the lay person to understand and follow. Is it possible for the authors to offer the readers a very simple end-result systems map they could follow and implement? Response 7: We have increased the font size in figures 2-5 to make them more readable. Unfortunately it is not possible to offer a simpler map for users to follow and implement, because the map is intended to represent the complexity of real world behaviour, which is not simple and linear. In the online version of the map we have offered various interactive options to help users navigate the complexity. Comment 8: Under limitations, the authors should write that this study involved individuals in Kisumu, Kenya and may not be generalizable to other communities, especially communities that are not in developing nations. Response 8: We have added a paragraph about this point under limitations (lines 650-654).
Reviewer 3 Report
Comments and Suggestions for Authors
The authors conducted an in-depth case study on solid waste management in Kisumu, Kenya. Utilizing strategic interview data, they extracted textual data, which was then analyzed through the COM-B framework. Through behavior-system mapping, they identified three subsystems and described the various actors and their cross-functional roles.
First, the study’s focus on a specific region highlights its urgency and significance, which is commendable.
Second, the use of high-value data derived from strategic interviews is a notable strength of this article.
Third, the authors’ systematic analysis and verification process significantly enhance the reliability of the findings, further contributing to the scientific robustness of the paper.
On the other hand, the data employed in this article presents opportunities for extended research using various textual analysis methods. Techniques such as co-occurrence frequency of words, centrality analysis, Euclidean distance, and network analysis could uncover additional actors and their roles. I look forward to seeing these approaches explored in future studies.
Author Response
Thank you very much for your review report. We appreciate your feedback and we did not identify any requested changes to the paper among your comments.
Round 2
Reviewer 2 Report
Comments and Suggestions for Authors
Second review: Behavioural systems mapping of solid waste management in Kisumu, Kenya
Comment 1 Title: This reviewer understands the desire to be concise with the title but the reader may be less inclined to read the article if it only is about developing a behavioral map. Why was this behavioral systems mapping undertaken? What about considering the inclusion of text in the last line in the abstract. “Behavioural systems mapping of solid waste management in Kisumu, Kenya to understand behavior changes that may reduce impacts of waste.”
Comment 2: Abstract: This reviewer understands the limitations of word counts in abstracts, but the current abstract now has the word or variations of the word “behavior” 15 times. Often readers only read the abstract and then do not read the article so having text in the methods section might not be read by a potential reader. The text on lines 14-15 and 18-19 is very similar. On line 25, the authors write that they identified six key feedback loops that contribute to poor waste outcomes. What are these? The current abstract tells the reader what the authors did (develop the mapping system) but the reader will want results. The text in lines 25-29 repeats much of the earlier text, again saying what the authors did and what the mapping can do and not what was learned for results. In the key words, the authors write communicable diseases but there is no mention of communicable diseases in the abstract. While having the information about Kisumu in lines 146-173 is helpful, it would be better to have less repetition in the abstract and tell the reader about Kisumu in the abstract. This text underscores the urgency and makes a better case for having the mapping system than just repeating the same text in the abstract.
Comment 3: The statement that the results can be complex for most readers is true. Any simplification for readers would be of value because the topic of waste management is a dire concern for the world.
Comment 4: The text in lines 325-326 includes this language, “The public waste management sub-system describes practices happening on the ground, such as generating, segregating, mixing, collecting, transporting, picking and burning waste.” This text is not in the abstract. Could the authors try to write the abstract without using the word “behavior” numerous times and see if there is sufficient word count to include the other rich and informative text in the study?
Comment 5: This reviewer understands the need to validate the map. The added text on lines 650 to 654 is helpful. This text and a version of the text in lines 645 to 647 could be at the end of the abstract. Lessons learned in this study that took place in a sub-Saharan African context may prove of value by revealing the similarities and dissimilarities to other countries. All countries now are dealing with waste.
Comment 6. Thank you.
Comment 7: Thank you for increasing the drawing.
Comment 8: Thank you. As written in Comment 5, it necessary to identify the lack of generalizability of this study but lessons can always be learned by seeing what is different and what is the same in different countries.
Author Response
Thank you very much for your second review report. We appreciate your suggestions and have amended the manuscript in line with your feedback. Please find our responses to specific comments below.
Comment 1 Title: This reviewer understands the desire to be concise with the title but the reader may be less inclined to read the article if it only is about developing a behavioral map. Why was this behavioral systems mapping undertaken? What about considering the inclusion of text in the last line in the abstract. “Behavioural systems mapping of solid waste management in Kisumu, Kenya to understand behavior changes that may reduce impacts of waste.”
Response 1: Thank you for the suggestion, we have updated the title with similar wording to: 'Behavioural systems mapping of solid waste management in Kisumu, Kenya to understand the role of behaviour in a health and sustainability problem' (lines 3-4)
Comment 2: Abstract: This reviewer understands the limitations of word counts in abstracts, but the current abstract now has the word or variations of the word “behavior” 15 times. Often readers only read the abstract and then do not read the article so having text in the methods section might not be read by a potential reader. The text on lines 14-15 and 18-19 is very similar. On line 25, the authors write that they identified six key feedback loops that contribute to poor waste outcomes. What are these? The current abstract tells the reader what the authors did (develop the mapping system) but the reader will want results. The text in lines 25-29 repeats much of the earlier text, again saying what the authors did and what the mapping can do and not what was learned for results. In the key words, the authors write communicable diseases but there is no mention of communicable diseases in the abstract. While having the information about Kisumu in lines 146-173 is helpful, it would be better to have less repetition in the abstract and tell the reader about Kisumu in the abstract. This text underscores the urgency and makes a better case for having the mapping system than just repeating the same text in the abstract.
Response 2: Thank you for the more detailed suggestions for improving the abstract. We have reduced the number of instances of 'behaviour' and 'behavioural' and reduced the amount of repetition. We have also removed communicable diseases from the keywords. We have added some more context about the waste problem in Kisumu ('open burning, illegal dumping and reliance on landfill', line 14) and explained the feedback loop results (lines 23-24). We agree that this will help the reader understand what was learned from the results and how they apply to the situation in Kisumu.
Comment 3: The statement that the results can be complex for most readers is true. Any simplification for readers would be of value because the topic of waste management is a dire concern for the world.
Response 3: We have further simplified the explanation of the results in the conclusion (lines 659-661) and included this in the abstract (lines 23-24).
Comment 4: The text in lines 325-326 includes this language, “The public waste management sub-system describes practices happening on the ground, such as generating, segregating, mixing, collecting, transporting, picking and burning waste.” This text is not in the abstract. Could the authors try to write the abstract without using the word “behavior” numerous times and see if there is sufficient word count to include the other rich and informative text in the study?
Response 4: We have reduced the number of times we mention 'behaviour' in the abstract and have included more details about the feedback loop results, including the handling and segregation of waste (lines 23-24).
Comment 5: This reviewer understands the need to validate the map. The added text on lines 650 to 654 is helpful. This text and a version of the text in lines 645 to 647 could be at the end of the abstract. Lessons learned in this study that took place in a sub-Saharan African context may prove of value by revealing the similarities and dissimilarities to other countries. All countries now are dealing with waste.
Response 5: We agree a version of this text should be included in the abstract and have added: 'Further research to validate and adapt this approach may extend the learnings to other countries and health and sustainability challenges.' (lines 27-28)